# Food from faeces: Evaluating the efficacy of scat DNA metabarcoding in dietary analyses

David Thuo[1,2]*, Elise Furlan[1], Femke Broekhuis[2,3], Joseph Kamau[4,5], Kyle Macdonald[6], Dianne M. Gleeson[1]

**1** Institute for Applied Ecology, University of Canberra, Bruce, Australian Capital Territory, Australia, **2** Kenya Wildlife Trust, Nairobi, Kenya, **3** Wildlife Conservation Research Unit, Department of Zoology, University of Oxford, Recanati-Kaplan Centre, Tubney, United Kingdom, **4** Molecular Biology Laboratory, Institute of Primate Research, Nairobi, Kenya, **5** Department of Biochemistry, University of Nairobi, Nairobi, Kenya, **6** National Zoo and Aquarium, Canberra, Yarralumla, Australian Capital Territory, Australia

* david.thuo@canberra.edu.au

**Data Availability Statement:** All relevant data are within the manuscript and its Supporting Information files, also the dataset (including the raw metabarcoding dataset) are available in the

## Abstract

Scat DNA metabarcoding is increasingly being used to track the feeding ecology of elusive wildlife species. This approach has greatly increased the resolution and detection success of prey items contained in scats when compared with other classical methods. However, there have been few studies that have systematically tested the applicability and reliability of this approach to study the diet of large felids species in the wild. Here we assessed the effectiveness of this approach in the cheetah *Acinonyx jubatus*. We tested how scat degradation, meal size, prey species consumed and feeding day (the day a particular prey was consumed) influenced prey DNA detection success in captive cheetahs. We demonstrated that it is possible to obtain diet information from 60-day old scats using genetic approaches, but the efficiency decreased over time. Probability of species-identification was highest for food items consumed one day prior to scat collection and the probability of being able to identify the species consumed increased with the proportion of the prey consumed. Detection success varied among prey species but not by individual cheetah. Identification of prey species using DNA detection methods from a single consumption event worked for samples collected between 8 and 72 hours post-feeding. Our approach confirms the utility of genetic approaches to identify prey species in scats and highlight the need to account for the systematic bias in results to control for possible scat degradation, feeding day, meal size and prey species consumed especially in the wild-collected scats.

## Introduction

Development of accurate methods to study the diet of terrestrial carnivores has been an active area of research and continues to attract increasing interest in conservation studies. Feeding patterns are a fundamental part of carnivore ecology and conservation [1]. Therefore, accurate inferences of breadth and diversity of feeding behaviour in the wild is required to understand their impacts on the ecosystem to develop reliable management programs of rare prey species and to predict potential human-wildlife conflicts [2–5]. However, it is often challenging to

Dryad Digital Repository: https://doi.org/10.5061/dryad.2z34tmpgs.

**Funding:** The authors received no specific funding for this work.

**Competing interests:** The authors have declared that no competing interests exist.

accurately infer carnivore diets because most terrestrial carnivores exist in relatively low numbers and are generally elusive and wide-ranging [6,7] and often opportunistic, thus making observational studies of diet logistically difficult, financially expensive and almost impossible under natural conditions [8].

DNA-based diet analyses of non-invasively collected samples, e.g scat DNA metabarcoding (sDNA metabarcoding) has been presented as a reliable alternative method [9–12]. This technique analyses DNA contained in scats collected from the wild using high-throughput sequencing using small, highly variable universal primers (barcodes) [13,14] to identify prey species. Relative to conventional dietary studies that typically rely on morphological identification of undigested remains in scats [9], sDNA metabarcoding has been shown to have higher sensitivity, greater taxonomic resolution and to be relatively cost-efficient [15,16]. In order to determine the reliability of sDNA metabarcoding, several controlled experimental studies have been conducted to examine the potential strengths and weaknesses. These studies have mainly scrutinized the specificity and sensitivity of PCR assays [17,18], library preparation and sequencing technologies [19–21], impact of environmental factors on scats [22], biological and physiological status of the defecator [22,23]. Few sDNA studies have empirically tested the effectiveness of sDNA metabarcoding in large felids, (but see [10]), and therefore drawing general conclusions from different taxa may introduce bias in result interpretation.

Prey DNA detectability in scat varies depending on both the prey species eaten and the predator species [24]. Thus, species-specific studies are needed to understand how biological, technical and environmental factors could affect the prey DNA signature recovered from a scat sample to inform optimal study design. Studies of captive animals with known diets allow sDNA methods to be trialed with the aim of maximizing prey detectability and identifying optimal designs for field studies [1,25].

The cheetah *Acinonyx jubatus* is Africa's most endangered large cat with the majority of remaining wild populations existing outside protected areas and hence prone to negative human interactions [25,26]. Cheetahs have large home ranges, are cryptic [27,28] and usually conceal their kills to minimize losses to other predators [29]. Consequently, monitoring of cheetah dietary habits using direct observation or carcasses can be time-consuming and expensive. Although cheetahs consume more pure muscle than bone and skin [30], prey items can be identified in cheetah scat samples [31,32], suggesting that sDNA metabarcoding has potential for wild cheetah dietary studies. Cheetah scats can persist in the field under dry environmental conditions for weeks and can easily be located at marking trees or using professionally-trained scent detection dogs [33]. However, obtaining freshly deposited cheetah scats in the wild is difficult, and it is not known how aging affects the ability to detect prey in cheetah scats.

The aim of this study was to analyse scats obtained from the captive cheetahs fed a known diet to address two questions (i) what is the length of time after consumption that prey DNA is detectable in fresh scats as a function of prey species and proportion of prey consumed, and (ii) how does the detection probability change over time in scats left outside to degrade. We discuss how these findings can be used to inform sDNA metabarcoding studies of wild cheetah diets.

## Materials and methods

### Feeding trials

We conducted a controlled feeding trial with two adult male cheetahs (Jura and Innis) between 2 November and 20 November 2017. The cheetahs are brothers born in 2013 and housed individually in outdoor enclosures at the National Zoo and Aquarium in Canberra, Australia.

During the study period, Jura and Innis were fed six prey species; horse *(Equus caballus)*, rabbit *(Oryctolagus cuniculus)*, deer *(Cervus spp)*, quail *(Coturnix Coturnix)*, chicken *(Gallus gallus)* and turkey *(Meleagris gallopavo)* in different proportions on different days (Table 1). Each day the selected prey items were weighed, placed in a bowl and fed to the individual cheetah. The cheetahs were fed once a day between 9 am and 11 am with total daily food intake varied based on cheetah body condition scores [34,35]. Jura weighed 53.9kg and was fed 1700g of food daily while Innis, weighed 50.4kg and was fed 1800g of food daily. To investigate the window of prey DNA detection in fresh scats (i.e. the number of days after consumption of a prey item that the prey was detectable in scats), Innis was fed once on quail hereafter referred to as spike diet, on day two of the experiment.

### Scat sampling

During the feeding experiment, scat samples from both cheetahs were collected daily except for days when the cheetah did not defecate. We collected a total of 16 and 10 fresh scats from Jura and Innis respectively. All fresh scats were placed in separate greaseproof paper bags and transported to the University of Canberra. For each scat, ~5 grams of material were subsampled on the day of deposit and stored at -20˚C. The remaining scats were then placed outside in an open field about 10 metres apart and exposed to natural weather to simulate wild conditions. Scats were individually labelled, and their location marked using 10" metal garden stakes. Each scat was then subsampled by removing ~5 grams of material on days 3, 5, 12, 15, 20, 27, 48 and 60 after being placed in the open. Not all scats survived to day 60 as some were eaten or removed, most likely by birds, foxes or insects. For subsampling, each scat was cut cross-sectionally using single-use sterilized surgical blade (Livingstone International, Australia) and material was taken from the upper, middle and lower surface of the cross-section.

In total, 203 subsamples were collected for DNA extraction. Daily weather data (temperature, rainfall and relative humidity) throughout the experiment was obtained from the nearest weather station (approximately 11 kilometres) to the open field site (Canberra Airport Station; Bureau of Meteorology, Australia 2018).

### Primers

We amplified the scat DNA using a previously published universal vertebrate primer set [17]. The primer set was selected based on taxonomical coverage and discrimination power. This set of primers has been demonstrated to have high-resolution capacity to identify the genus and species across a wide range of vertebrate taxa [17]. This primer pair amplifies an ~100 bp fragment of the V5 loop of mitochondrial 12S rRNA gene (Table 2).

### DNA extraction and PCR amplification

Approximately 0.1–0.2g of the material was removed from each scat subsample and DNA was extracted using the Invitrogen ChargeSwitch® Forensic DNA Purification Kit (Invitrogen™ Life Technologies, USA) following the manufacturer's instructions and using overnight digestion at 55˚C rocking at 850rpm in a thermomixer. Samples were extracted in batches of 23 including a negative control in which no sample was added. In order to assess the amplification efficiency and inhibition, all extracts were diluted to 1/10 and 1/100 and used along with undiluted aliquot during qualitative PCR (qPCR) amplification. All qPCR reactions were carried out in 25µl consisting of final concentration of: 0.20 µl of AmpliTaq Gold DNA Polymerase (Applied Biosystems, USA), 2.5µl of GeneAmp 10× Gold Buffer (Applied Biosystems, USA), 2µl of MgCl2 (25 mmol/L; Applied Biosystems, USA), 0.2µl UltraPure BSA (50 mg/ml; Invitrogen), 0.65 µl of GeneAmp dNTP Blend (10 mmol/L; Applied Biosystems, USA), 0.6 µl

**Table 1. List of prey species (and proportions) fed to cheetahs each day during the captive feeding experiment.**

| Day, month and year | Cheetah ID | Prey species 1 | Prey species2 | Prey species 3 |
|---|---|---|---|---|
| 03.11.2017 | Jura | Deer (0.47) | Chicken (0.18) | Rabbit (0.35) |
| 04.11.2017 | Jura | Deer (0.47) | Chicken (0.29) | Rabbit (0.24) |
| 05.11.2017 | Jura | Deer(0.82) | Chicken (0.18) | - |
| 06.11.2017 | Innis | Deer (0.56) | Horse (0.27) | Chicken (0.17) |
| | Jura | Deer (0.82) | Chicken (0.18) | - |
| 07.11.2017 | Innis | Horse (0.61) | Turkey (0.06) | Chicken (0.33) |
| | Jura | Deer (0.82) | Chicken (0.18) | - |
| 08.11.2017 | Innis | Deer (0.56) | Rabbit (0.6) | Quail (0.38) |
| | Jura | Deer (0.88) | Chicken (0.6) | Rabbit (0.6) |
| 09.11.2017 | Innis | Horse (0.11) | Rabbit (0.6) | Chicken(0.83) |
| | Jura | Deer (0.88) | Horse (0.12) | - |
| 10.11.2017 | Innis | Rabbit (0.17) | Chicken (0.83) | - |
| | Jura | Deer (0.88) | Chicken (0.12) | - |
| 11.11.2017 | Innis | Horse (0.33) | Chicken (0.67) | - |
| | Jura | Deer (0.88) | Chicken (0.12) | - |
| 12.11.2017 | Innis | Deer (0.89) | Chicken (0.11) | - |
| | Jura | Deer (0.88) | Chicken (0.12) | - |
| 13.11.2017 | Innis | Deer (1.0) | - | - |
| | Jura | Deer (0.88) | Chicken (0.12) | - |
| 14.11.2017 | Innis | Rabbit (0.22) | Chicken (0.78) | - |
| | Jura | Deer (0.88) | Chicken (0.12) | - |
| 15.11.2017 | Innis | Rabbit (0.22) | chicken (0.78) | - |
| | Jura | Deer (0.88) | Chicken (0.12) | - |
| 16.11.2017 | Jura | Deer (0.88) | Chicken (0.12) | - |
| 17.11.2017 | Jura | Deer (0.88) | Chicken (0.12) | - |
| 18.11.2017 | Jura | Deer (0.88) | Chicken (0.12) | - |
| 19.11.2017 | Jura | Deer (0.88) | Chicken (0.12) | - |

SYBR Green I Nucleic Acid Gel Stain (5X; Invitrogen), 1μl of forward and reverse primer (10 μmol/L), and 3μl of template DNA and made to volume with DEPC-treated water (Invitrogen™ Life Technologies, USA). Each qPCR was run using a Bio-Rad CFX96 Real-Time PCR System (Bio-Rad Laboratories, Hercules, USA) under the following conditions: initial activation at 95˚C for 5 min, followed by 45 cycles of 95˚C for 30 sec, 57˚C for 30 sec, and 72˚C for 2 min and a final extension of 10 min at 72˚C and a melting curve with a stepwise increase of 0.1˚C/5 s from 60 to 95˚C completed the reaction. The PCR set-ups were conducted in a dedicated trace DNA laboratory at the University of Canberra to minimise the risk of contamination. The DNA dilution with the highest relative proportion of starting template (determined by $C_q$ values) was selected for subsequent metabarcoding using fusion-tagged primers. All negative control samples that showed positive amplification were included in the high-throughput sequencing library preparation.

**Table 2. Details of the primer sequences used in the study.**

| Primer name | Primer sequence (5´ - 3) | Product size | References |
|---|---|---|---|
| 12SV5F | TAGAACAGGCTCCTCTAG | ~100bp | [17] |
| 12SV5R | TTAGATACCCCACTATGC | ~100bp | [17] |

## Library preparation and high-throughput sequencing

A single step PCR with fusion-tagged primers was used to amplify the barcoding sequence and add technical sequences required for high-throughput sequencing. Forward fusion-tagged primers consisted of the P5 sequencing adaptor, a custom forward sequencing primer, a 7 bp Multiplex Identification (MID) tag, and the forward 12SV5 primer. Reverse fusion-tagged primer contained the P7 sequencing adaptor, a custom reverse sequencing primer, a 7 bp MID-tag, and the reverse 12SV5 primer. To minimize cross-contamination, no primer-MID combination had been previously used, nor were combination re-used. Triplicate PCRs were run for each sample using the reaction conditions and thermal cycling profile described previously. Based on the average quantitation cycle value (Cq values) of each sample, amplicon libraries of 8–10 samples were pooled using equal volumes of each PCR replicate to produce a single DNA library. All negative controls were pulled together into a single unique library. Tagged amplicons were purified (to remove excess fusion-tagged primers and primer dimers) using Agencourt™ AMPure™ XP Beads (Beckman Coulter, Brea, CA, USA) in a 1.2 volume ratio relative to the amplicon pool.

The size and concentration of the amplicons of each pool were estimated by electrophoresis on 2% agarose gel stained with SYBR safe (Invitrogen™ Life Technologies, USA) and Nano-Drop® ND-1000 spectrophotometer (Thermo Fisher Scientific, Waltham, MA, USA). Based on pools equimolar concentration, amplicons were combined to produce a single super pool. The super pool was constructed by combining approximately equal amplicon copy numbers from each initial pool (i.e., considering the number of samples combined during the first pooling step and the amplicon size). A total of 209 uniquely labelled libraries from this study (i.e., 193 and 18 libraries originating from scat DNA and negative control samples, respectively) were included in the final superpool. The resultant library was purified as described above. All sequencing for the 209 libraries was performed using Illumina MiSeq® with the Version 2 reagent 1x200 bp reagent kit at the Ramaciotti Centre for Genomics (University of New South Wales).

## Bioinformatics data processing

The technical sequences (i.e. sequencing adaptors and primers) from the sequencing reads were trimmed using Trimmomatic v.0.36 [36]. Low-quality bases (Q-score < 30) at the end of the sequencing reads were filtered out and a sliding window of 4-bases was used to trim reads when the average quality per base was below 15. The OBITOOLS software [37] was used for subsequent filtering of the sequences following the general workflow described by De Barba et al (2014)The OBITOOLS *ngsfilter* and OBIGREP scripts were used to assign sequences records to the corresponding sample combination and remove any sequences shorter than 80 base pairs in length and with abundance below 10 [10], as they could potentially be sequencing errors and/or chimaeras. OBICLEAN and OBIGREP scripts were used to remove PCR and sequencing errors. The ECOTAG script was used to assign the sequences to their corresponding taxonomic information using a reference database built using the standard vertebrate sequences from the EMBL data repository (release 138; https://www.embl.de/) and a 12SV5 custom reference database built specifically for our target species: cheetah, horse, rabbit, deer, quail, chicken and turkey. ECOTAG output files were imported into in R version 3.5.1 (https://www.R-project.org/) for further filtering and statistical analyses using tidyverse [38], lubridate [39], JAGS [40] and jagsUI [41]

During ECOTAG, some sequences were assigned to higher taxonomic ranks than the species level. Since all the species in our feeding experiment were known and all sequences assigned to higher taxonomic ranks had variant sequences assigned to species level with a

higher occurrence, these incorrect assignments were reassigned to the species species. Unclear taxonomic assignments were either modified or corrected based on the relative sequence abundance, the sequence information and the prior knowledge of the expected species. For example, all sequences assigned to the Felidae family were combined into a single species level assignment *Acinonyx jubatus*, as it is likely they are from the cheetah. Additionally, all sequences assigned to the Leporidae Family were reassigned to *Oryctolagus cuniculus* species, all sequences assigned to Equidae family were reassigned to *Equus caballus* species and those assigned to Cervidae family combined into Cervus species. All other sequences from non-target species (not from cheetah or prey species in the cheetah feeding experiment) or without a taxonomic assignment were excluded from downstream analyses.

## Data analysis

Due to differences in the sequencing depth among samples, the ECOTAG output data was transformed into binary data on the presence or absence of each prey species in each scat subsample. A prey species was considered to be present in a scat subsample if its sequence reads were detected but were missing or less than ten in the corresponding negative control.

Quail (spike diet) was detected in scats up to three days post feeding. Based on this knowledge, we excluded from analysis scats that were collected in the first three days of the feeding trial as we did not know what the cheetahs had been fed in the days prior to the start of the experiment. This resulted in, one prey species (Turkey *Meleagris gallopavo*) being excluded from the analysis because it was only fed to one cheetah within the first three days.

For each scat we had data on what the cheetah had consumed on the day of defaecation and for 3 consecutive days prior to scat collection, and for each subsample taken from each scat we had data on the presence or absence of prey species in that subsample. We modelled the presence of prey species in each scat subsample as a function of the proportion of each prey type that was fed to a cheetah in each of the previous three days, the number of days since a scat was defecated (degradation days) and the individual cheetah. The response variable was detection of prey species in a subsample from scat $i$ on degradation day $j$, $Y_{s,ij}$, coded as $Y_{s,ij} = 0$ (if the prey species was not detected) or $Y_{s,ij} = 1$ (if the prey species was detected). We modelled the probability of detection, $p_{s,ij}$, as a function of 6 fixed-effect covariates: an intercept term; the proportion of prey species fed to the cheetah on the day of defecation and on each of the three days prior to that, the number of days after defecation that the scat was subsampled (degradation days) and the individual cheetah. We also included a random effect term $\alpha$ with a different value for each scat that accounted for repeated measures in the data with multiple subsamples taken from each scat. Our model was:

$$Y_{s,ij} \sim \text{Bernoulli}(p_{s,ij}) \tag{1}$$

$$\text{Logit}(P_{sij} = \beta_{0,s} + \beta_{1,s} * \text{pr0}_{,s,i} + \beta_{2,s} * \text{pr1}_{,s,i} + \beta_{3,s} * \text{pr2}_{,s,i} + \beta_{4,s} * \text{pr3}_{,s,i} + \beta_5$$
$$* \text{degredation day}_j + \beta_6 * \text{cheetah} + \alpha_i) \tag{2}$$

Where $i$ indexes scats (1–26), $j$ indexes degradation days (1–60) and $s$ indexes prey species (1–5). $\beta_{0,s}$ is the baseline probability of detection for prey species s, $\beta_{1,s}$–$\beta_{4,s}$ are parameters that describe how the probability of detection depends on the proportion of each prey species eaten on the day of defaecation ($\beta_{1,s}$) or in the preceding three days ($\beta_{2,s}$–$\beta_{4,s}$), $\beta_5$ is a parameter that estimates how probability of detection changes as a function of scat degradation day, $\beta_6$ estimate the effect cheetah has on detection, and $\alpha$ is a random-effect term that allows a different overall detection probability for each scat.

We fit the models using Bayesian methods and estimated the posterior distribution for all parameters using Markov Chain Monte Carlo (MCMC) implemented in JAGS [40] within the package jagsUI Version 1.5.0 [41] in R environment [42]. The β and $\alpha_i$ parameters were modelled hierarchically, assuming these were drawn from normal distribution with means and variance estimated from the data for the β parameters, and mean zero and variance estimated from the data for the $\alpha_i$ parameters. We used non-informative priors for the means (mean 0 and variance 100) and variances (uniform prior in the range 0–10 on the standard deviation). The models were run using three Markov chains of 20,000 iterations after a burn-in of 5000 iterations until all parameters were judged to have converged based on Gelman-Rubin statistic (Rhat statistic), for which all values were <1.1 [43]. To assure full reproducibility of our data analyses we have provided all datasets and workflow as supporting information (S1, S2, S3 and S4 Files). The raw metabarcoding data and R code used for the analysis are available in the Dryad Digital Repository https://doi.org/10.5061/dryad.2z34tmpgs) [44]

## Results

### Weather

During the study period, the study site received rain on 37 days for a total of 290mm. The temperature ranged from 2.5˚C to 40.6˚C with an average temperature of 26.9˚C. The average minimum temperature over the entire study period was 12.8˚C and the average maximum was 28˚C. Relative humidity ranged from 11.7% - 100%, with an average relative humidity of 60.7%.

### Bioinformatics

After quality filtering and removal of chimaeras, a total of 15,306,489 sequence reads were obtained of which 12,254,953 reads (80%) included perfectly matching MID. The remaining 20% either did not have MID or had MID tag with numerous mismatches to be reliably assigned. Overall, the quality of run was high (PhredQ30 score $\geq$ 90.53, error 1.04 ± 0.03). As expected, more than half of the sequence reads (54%) were assigned to the consumer (cheetah), while 33% were assigned to prey items and the remaining 13% of the total sequence reads being assigned to other. These findings are consistent with the literature [14,19,45], this is due to the high number of epithelial cells/cells of the intestinal mucosa from the defecating animal and probable prey DNA decay due to digestion process [46]. Two of the extraction controls that had shown positive amplification did not result in assignment during ECOTAG process possibly because the initial positive amplification was due to the 12SV5 primers amplifying non-target (e.g. microbial) DNA or due to primer dimer formations.

### Diet

The number of days since consumption and proportion of prey fed strongly influenced prey DNA detection in the cheetah scats. Averaged across all prey species, there was a positive relationship between the probability of detection per proportion of prey consumed, although this effect was weak on day 0 (the day of consumption), peaked on day 1 (the day after consumption) and then declined in the following two days (Fig 1 and Table 3).

Nevertheless, these relationships also appeared to vary depending on the prey species consumed (Fig 2): chicken, deer and horse were more readily detected on the day of consumption compared to quail and rabbit, while horse was difficult to detect after day one.

Degradation day (number of days the scat was exposed to the environment) was weakly negatively associated with detection probability for scats exposed to natural conditions for up

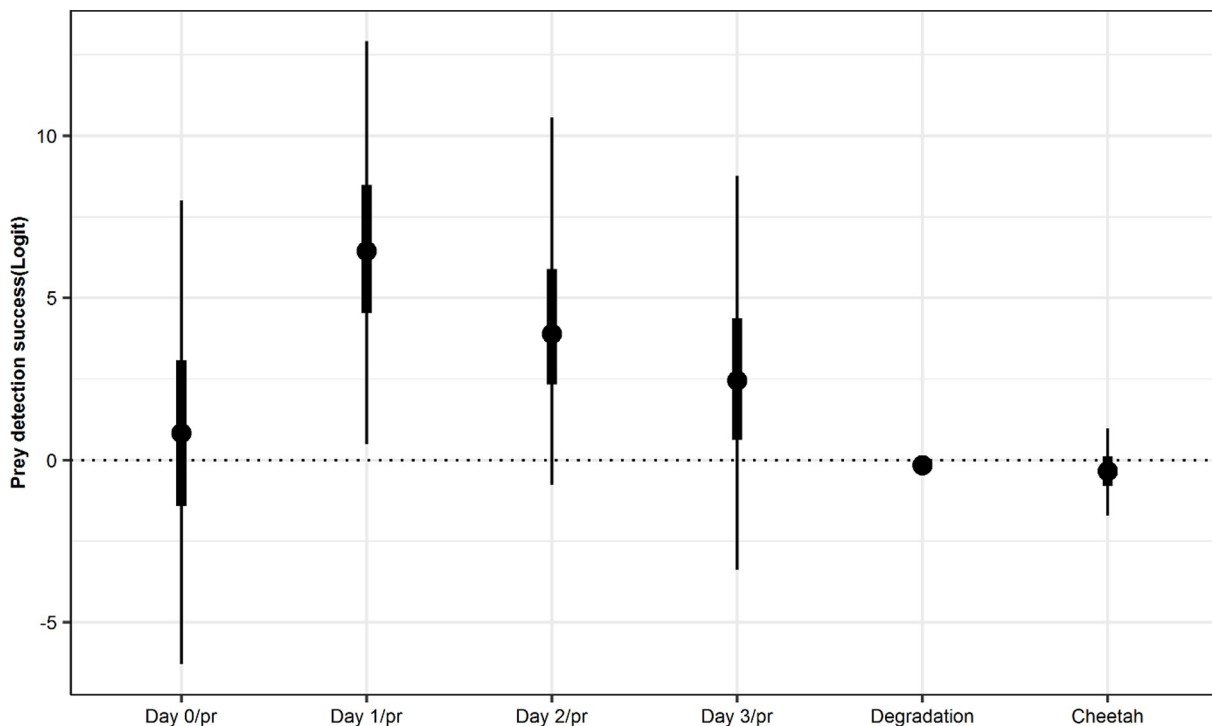

**Fig 1. The relative success of prey DNA detection on a given day after feeding (according to the proportion of prey consumed), degradation day and individual cheetah.** The points are the posterior means and the bold and thin lines represent the 50% and 95% credible intervals around the means respectively.

to 60 days (Fig 1 and Table 3). There was no clear difference between individual cheetahs in the probability of prey detection (Fig 1 and Table 3).

Detectability varied among prey species indicating the need to account for this bias when evaluating the cheetah diet (Fig 2). Chicken showed the highest probability of detection (75% SD: 0.18) while quail and rabbit (13% SD: 0.25 and 4% SD:0.06) showed the least probability of detection in day zero respectively i.e. the same day the cheetah was fed. The probability of detection declined after day one for horse and after day two for chicken and rabbit. Quail and deer showed no clear differences in detection probabilities among days.

Using the raw dataset to evaluate the relationship between meal sizes and the probability of prey detection, the results supported a positive correlation, where the probability of detection increased with increase in meal size (Fig 3).

**Table 3. Posterior summary of the model.**

| Parameters | Posterior means | Standard deviation | 95% Credible interval | |
|---|---|---|---|---|
| | | | Lower limit | Upper limit |
| Day 0/pr fed | 0.01 | 3.24 | -6.72 | 6.51 |
| Day 1/pr fed | 4.43 | 2.55 | -0.56 | 9.85 |
| Day 2/pr fed | 1.82 | 1.69 | -1.32 | 5.67 |
| Day 3/pr fed | 1.04 | 2.66 | -4.30 | 6.58 |
| Degradation | -0.16 | 0.09 | -0.35 | 0.02 |
| Cheetah | -1.19 | 0.64 | -2.51 | 0.05 |

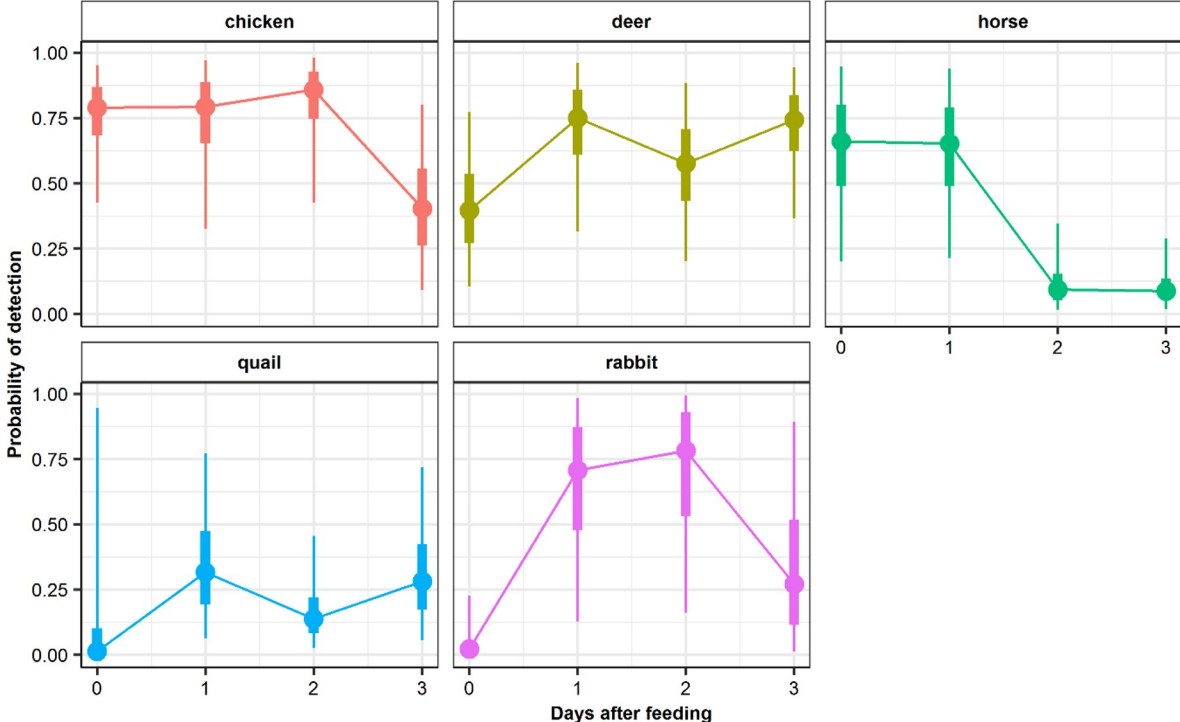

**Fig 2. Estimates of mean detection probability of each prey species in scat samples relative to time since feeding.** The bold and thin lines represent the 50% and 95% credible intervals around the means.

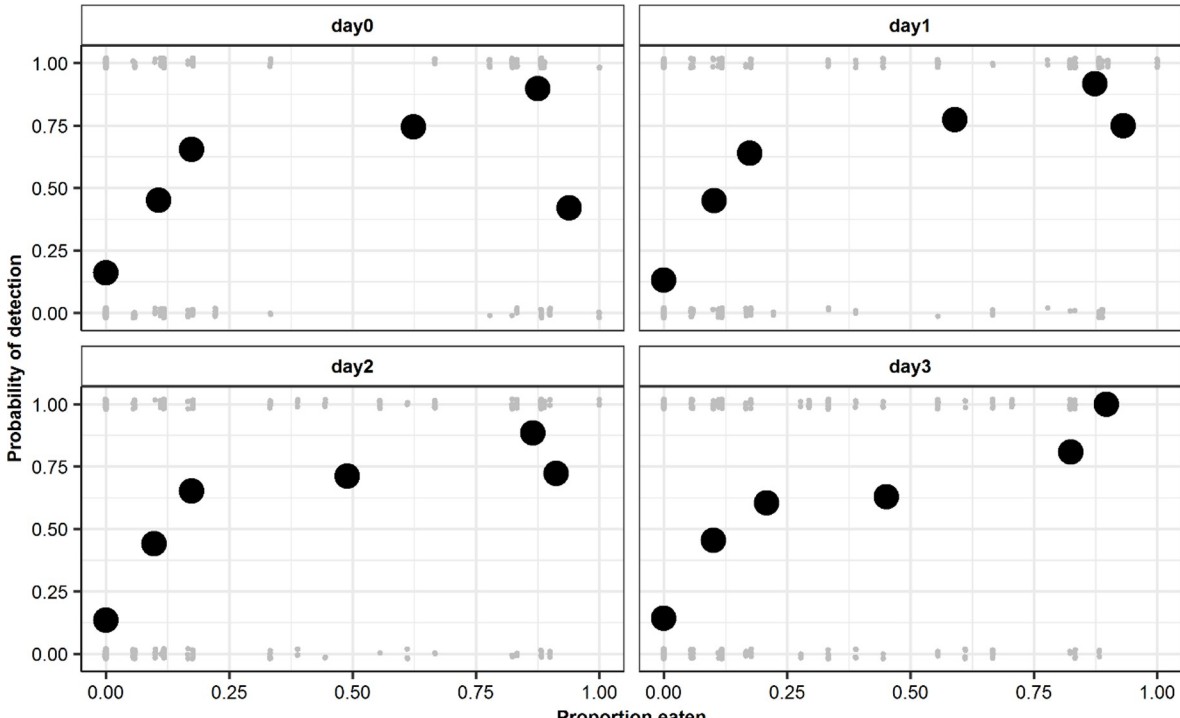

**Fig 3. Probability of prey detection as a function of meal size.** The grey dots at 0.00 and 1.00 indicate absence or presence of detection of prey items respectively, and the black circles shows the proportion of prey detection relative to proportion fed.

The initial detection of the spike diet was possible within 24 hours post feeding (minimum gut transition time) and could still be detected until 72 hours (maximum gut transition time). We did not detect the spike diet in scats collected after 72 hours.

## Discussion

Our results demonstrate that scat DNA metabarcoding provides a sensitive method of prey detection in cheetah scats. All the prey species fed to the cheetahs during the feeding experiment were detected and therefore show the potential utility of this approach in field studies where prior information on diet of cheetah is not known. However, this study did show that prey DNA detection was influenced by different variables namely feeding day, degradation (scat age), consumed prey species, and the meal size consumed by the cheetah, which also need to be considered when making interpretations from field samples.

Our hierarchical model showed that prey detection was greatly influenced by the amount of time since being fed. Food items consumed by the cheetahs one day prior to scat collection had the strongest positive effect while a food item consumed the same day the scat was collected had the least influence on prey DNA detection. This trend follows expectations as more than 50% gastric emptying in most mammals happen within 40 hours [47]. Moreover, it is also likely that cheetahs have high digestibility efficiency similar to that observed in domestic cats [48,49]. If this holds true, the errors or bias introduced by feeding day could affect prey inferences, especially when diagnosing rare prey species or economically valuable prey e.g livestock which may not be a common prey species in the wild. Given that scat collection in the wild is not sequent and it is difficult to determine the time since the prey species was consumed, drawing a conclusion from scat DNA metabarcoding data by only estimating the frequency of occurrence could bias the diet estimates. Frequency of occurrence summarizes the proportion of samples containing a certain diet item, hence false negatives could arise if a scat was collected either too soon or too late after the consumption of prey [8,50]. These findings highlight the need for a more stringent scat collection protocol when planning for wild cheetah dietary studies perhaps by conducting an intensive scat collection within a short time period or by using a large number of scat samples collected over time.

We assessed whether degradation days (number of days a scat was exposed to the natural environment) had a significant impact on prey DNA detection on cheetah scats. Overall, this parameter showed a negative effect on prey detection. Similar results were reported earlier in scat analysis studies showing that detection of prey DNA is higher in fresh than in old scats [22,23,51]. However, contrary to the short maximum degradation time reported in the previous studies (e.g. 5–7 day old scats in Steller lion *Eumetopias jubatus* and 5 days old scat in carrion crows *Corvus corone*), our results indicate that prey detection is possible in cheetah scats that have been exposed to the open environment for up to 60 days under spring-summer conditions which have been shown to reduce prey detection success [23]. These results could indicate a potential species-specific food DNA detection success in old scats. This observation holds true as the diet of extinct ground sloth *Nothrotheriops shastensis* has been successfully inferred from fossilized scats [52]. During the degradation experiment, some samples were completely eaten or removed from the study site presumably by birds, foxes and/or small mammals, this is particularly relevant for field biologists planning a scat collection expedition as this would potentially affect the sample sizes.

The prey species consumed by the predators are recognized as an important consideration in scats dietary analysis and have been shown to influence the detectability of food DNA in scats [53]. Tissue composition and amount of DNA per gram of tissue vary across prey species hence some tissues are easy to digest and detect in scats [24]. Similarly, our study showed

variation in probabilities of detection among prey species. We found that detection success of chicken and horse was higher than that of deer, rabbit and quail. Of interest, our results showed that it is nearly impossible to detect some prey species on the same day they were consumed while it is highly feasible for others (Fig 3). The intuitive explanation is that the chicken and horse body parts fed to the cheetahs had high digestibility and contained high protein and lipid content and therefore could have reduced mitochondrial DNA decay during digestion. Thomas et al. (2014) in a feeding trial on harbour seals showed that fish with high protein levels tends to be overrepresented during diet recovery in scats. Other alternative factors that could explain our finding includes the meal sizes and frequency of feeding of a particular food item within the study period or they had high amount of bones and hair which may have increased their detection rates [54].

Estimate of prey DNA detection window from the spike diet results showed that the maximum passage time is 3 days post-feeding after which the spike diet DNA could no longer be detected in the scats. However, we could not explicitly determine the minimum passage time as the initial scat after feeding the cheetah on the spike diet was defecated at night and the exact time of defecation was therefore unknown. Consequently, we estimated the minimum passage time to be 8–22 hours post feeding. Although this conclusion is based on one spike diet, these findings were supported by the species-specific prey detection in our model that showed the probability of detection depends partly on the prey species with some species being detectable sooner after feeding and some being possibly detectable after 3–4 days (Fig 3). Maximum and minimum passage time in vertebrates is known to vary depending on diet composition, sex, physiological and satiation status of the consumer [23,48,55]. For cheetahs, gut transition time appears to be within the range of a few hours after feeding up to several days, meaning that a sample collected in the wild could potentially provide information on the cheetah's diet over the past 4 days. However, a lack of detection of a potential prey species may not necessarily mean its absence as food item, but possibly a failure to sample within the detection window.

The meal size can greatly influence the estimation of trophic ecology as large meals tend to have high detection rates as well as longer detection time span compared to small meals [56–58]. In our study, there was a positive relationship between meal sizes and the probability of prey detection. However, the relationship was also dependent on the feeding day, with the proportion of food consumed one day prior to scat collection having the highest positive effect on the detection, implying that the detection rate increases when a large meal size is consumed one day before a scat is collected (Fig 2).

We also showed that for 50% detection probability of prey in a scat, the prey item should have constituted approximately 20% of the cheetah's total daily consumed diet which in our study was approximately 300 grams. If these results hold true then this approach may be adequate in dietary studies of the wild cheetah as the maximum rate of consumption for wild cheetahs is estimated as 5.5 kg/day [59] implying a higher probability of prey detection per scat.

The plausible explanation for the uncertainty around the effects of the consumers (cheetahs) on prey detection is that the number of participating animals in our feeding experiment was small and biased towards males. To accurately account for this bias, further research is needed to explore the effects of sex and age by potentially using more cheetahs of different age groups. This is likely to be of particular importance as male cheetahs in the wild frequently occurs in coalitions and are larger than solitary females hence they kill larger prey [5,60]. Based on this, our hypothesis is that cheetah's sex and age may also affect prey DNA detection, with detection rates being higher for males as their meal size will likely be larger than that of

females and, consequently, might result in a higher quantity of prey mitochondrial DNA in scats.

In summary, scat DNA metabarcoding provides an efficient and accurate non-invasive tool to robustly assess the diet of cheetahs, but there are several confounding factors that should be considered when designing an optimal cheetah diet study. Our finding showed that the majority of sequence reads will emanate from the consumer and this could potentially reduce the prey information, therefore we recommend the use of blocking primers [61] to prevent the amplification of cheetah DNA templates. In addition, factors such as the meal size, prey species and the feeding day may drastically affect prey detection rates and thus, the inferences drawn from scat metabarcoding data may over or underestimate the prey breadth and diversity. To circumvent these limitations, we recommend the development of correction factors that would simulate field setup to maximise the usability of this approach.

## Supporting information

**S1 File. Cheetah feeding metadata.**
(CSV)

**S2 File. A file containing the summarised metabarcoding data.**
(CSV)

**S3 File. File containing the concatenated dataset- feeding and metabarcording datasets.**
(CSV)

**S4 File. Weather data collected during the study period.**
(CSV)

**S5 File. R code used to analyse the datasets.**
(R)

## Acknowledgments

The authors thank the National Zoo and Aquarium management and staff for permitting us to conduct this study in their facility and for assistance in scats collection. We particularly thank Richard Duncan for statistical advice and commentary on the manuscript, and two anonymous reviewers for comments/suggestions that greatly improved the previous version of the manuscript. We are indebted to Jonas Bylemans for his helpful guidance in wet and dry laboratories.

## Author Contributions

**Conceptualization:** David Thuo, Elise Furlan, Joseph Kamau, Dianne M. Gleeson.

**Data curation:** David Thuo, Elise Furlan, Femke Broekhuis, Kyle Macdonald.

**Formal analysis:** David Thuo, Femke Broekhuis.

**Funding acquisition:** Dianne M. Gleeson.

**Methodology:** David Thuo, Elise Furlan, Femke Broekhuis, Joseph Kamau, Kyle Macdonald, Dianne M. Gleeson.

**Project administration:** David Thuo.

**Resources:** Kyle Macdonald, Dianne M. Gleeson.

**Supervision:** Elise Furlan, Femke Broekhuis, Joseph Kamau, Dianne M. Gleeson.

**Validation:** Joseph Kamau, Kyle Macdonald.

**Visualization:** David Thuo.

**Writing – original draft:** David Thuo, Joseph Kamau, Dianne M. Gleeson.

**Writing – review & editing:** David Thuo, Elise Furlan, Femke Broekhuis, Joseph Kamau, Kyle Macdonald, Dianne M. Gleeson.

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
