## [Decision Letter · Decision Letter 0]

11 Sep 2019

PONE-D-19-20752

Food from faeces: evaluating the efficacy of scat DNA metabarcoding in dietary analyses

PLOS ONE

Dear mr Thuo,

Thank you for submitting your manuscript to PLOS ONE. After careful consideration, we feel that it has merit but does not fully meet PLOS ONE’s publication criteria as it currently stands. Therefore, we invite you to submit a revised version of the manuscript that addresses the points raised during the review process.

I got the recommendations and comments from an expert reviewer on the field. The reviewer agreed that the manuscript is technically sound and the data support the conclusions.However, lack of the explanation in Methods and Results sections were suggested by the reviewer, especially for statistical analysis and supplemental materials. I totally share their comments. Therefore, I can invite you to submit a revised version of the manuscript that addresses the points raised by the reviewer.

We would appreciate receiving your revised manuscript by Oct 26 2019 11:59PM. To enhance the reproducibility of your results, we recommend that if applicable you deposit your laboratory protocols in protocols.io, where a protocol can be assigned its own identifier (DOI) such that it can be cited independently in the future. For instructions see: http://journals.plos.org/plosone/s/submission-guidelines#loc-laboratory-protocols

We look forward to receiving your revised manuscript.

Kind regards,

Hideyuki Doi

Academic Editor

PLOS ONE

Journal Requirements:

1. We note that you are reporting an analysis of a microarray, next-generation sequencing, or deep sequencing data set. PLOS requires that authors comply with field-specific standards for preparation, recording, and deposition of data in repositories appropriate to their field. Please upload these data to a stable, public repository (such as ArrayExpress, Gene Expression Omnibus (GEO), DNA Data Bank of Japan (DDBJ), NCBI GenBank, NCBI Sequence Read Archive, or EMBL Nucleotide Sequence Database (ENA)). In your revised cover letter, please provide the relevant accession numbers that may be used to access these data. For a full list of recommended repositories, see http://journals.plos.org/plosone/s/data-availability#loc-omics or http://journals.plos.org/plosone/s/data-availability#loc-sequencing.

Additional Editor Comments (if provided):

I got the recommendations and comments from an expert reviewer on the field. The reviewer agreed that the manuscript is technically sound and the data support the conclusions.However, lack of the explanation in Methods and Results sections were suggested by the reviewer, especially for statistical analysis and supplemental materials. I totally share their comments. Therefore, I can invite you to submit a revised version of the manuscript that addresses the points raised by the reviewer.

Reviewers' comments:

Reviewer's Responses to Questions

**Comments to the Author**

1. Is the manuscript technically sound, and do the data support the conclusions?

Reviewer #1: Partly

2. Has the statistical analysis been performed appropriately and rigorously? 

Reviewer #1: No

3. Have the authors made all data underlying the findings in their manuscript fully available?

Reviewer #1: No

4. Is the manuscript presented in an intelligible fashion and written in standard English?

Reviewer #1: No

5. Review Comments to the Author

Reviewer #1: Summary:

The manuscript describes a feeding experiment carried out with two cheetahs, which were fed varying meat proportions of five (four) species. Defecations of the cheetahs were collected, subsamples taken and then exposed to natural conditions up to 60 days. During this time, scat subsamples were again taken. The aim was to test for the detectability of different prey species consumed at different quantities over time for future scat collection and subsequent molecular analysis in the wild.

Major comments:

The authors make a great case for why this experiment was necessary and how future field studies will benefit from the obtained results. Feeding experiments on large mammals are always hard to carry out because of limited individuals available and specific requirements to minimize stress of the animals. My main concern with this manuscript is twofold: on the one hand, materials and methods and the raw data uploaded as supporting information do not contain all the information necessary to fully comprehend the analyses. On the other hand, I honestly doubt that the feeding regime applied in this experiment permits all described analyses and conclusions. That being said, I would like to emphasize that such a trial is definitely useful prior to a large field sampling campaign even though it does not necessarily permit statistically robust answers to all questions raised in the introduction.

Supporting Information

Unfortunately, the manuscript and the Supporting Information do not enable the reader to combine data on the consumed diet with the prey detections. Especially for a situation where diet three days prior to defecation and prey detection up to 60 days after defecation plays a role, it would be great to have this information together in one dataset. Additionally, a legend describing the info in the Supplementary file columns would be very useful.

Some entries in the supplementary table are missing; for example, quail C.day 3, D.day 1 and 3. I am assuming that these were removed because of quail contaminations in the negative controls?! It would be great to see which samples had to be removed and why (contamination, scat consumed by other animals).

The dataset also does not clearly indicate which part of it was used in the final analysis, and which was not.

Statistic Analyses:

As the two cheetahs were offered different total amounts of prey, it would make more sense to use proportions (e.g. quail was 38% of total daily consumption) in all statistical analyses compared to absolute amount of prey consumed. The wording of the manuscript is not always clear; proportions are displayed in Table 1 but often in the text “amount” is used and in line 248 “kg” is mentioned as unit.

Results and Materials and Methods section do not fit well together i.e. it is not explained in Materials and Methods, how Figures 3 and 4 and the corresponding text were derived. My understanding is that Figure 3 and 4 and the corresponding text are based on detections in fresh scat samples (N = 26), but this might not be the case.

In my honest opinion, the analysis presented in the second part of the results section (Figure 3 and 4) are not appropriate under the feeding regime applied in this experiment.

Regarding the differences in detectability between prey species:

• For this analysis only fresh collected scats should be used, as the model presented in Figure 2 shows a negative influence of time since defecation on detection success. If subsamples collected after considerable time are used for this analysis, DNA degradation, environmental conditions and location of the subsample taken from the scat bias detection rates.

• Chicken was very frequently fed to both cheetahs. Therefore, a high detection rate of chicken in scats collected at day 0 after feeding is not surprising, as the chicken signal could easily stem from a meal consumed on the previous day.

Regarding the probability of detection as a function of meal size:

• Based on the feeding regime presented in Table 1, large “meals” (i.e. a prey species consumed in a large quantity) were not switched often. For example, Jura consumed a large meal of deer more than 10 times. Therefore, an increasing detection probability with increasing meal size across days 0 to 3 potentially stems from the same species being fed on consecutive days (or with just one day break) to the same animal.

For analyzing the effect of prey identity and detectability I would suggest using only results from the 26 fresh scats and omit all detections from the dataset where a species was consumed more than once during the 0-3 days time window, thus, reducing the analysis to the rarely consumed prey species like rabbit and quail. This would reduce the analytic power, but at least the nature of the feeding regime would not mask the actual effects.

For analyzing the effect of DNA degradation, I would suggest using scats produced by Jura during a constant “high deer, low chicken” diet and assess the detection probability of both the small meal and the large meal over time.

The Materials and Methods section or the Results section would benefit from a short description of the contamination levels found in the extraction controls and the amount of detections that had to be removed from the dataset prior to analysis because of contamination.

Minor comments:

Line 38: remove “This approach”

Lines 45-53: To some extent large carnivores are often opportunistic; it would be good to include this aspect here in the introduction.

Line 62: missing space after efficient.

Line 84: missing space between citation numbers

Line 95, Line 105: “amount consumed” as the two cheetahs were fed different daily amounts, I would suggest changing this into “proportion” and if necessary, changing the statistical analysis accordingly.

Lines 108-110: is it correct that the less weighting individual was fed larger meals?

Line 112: In my opinion, Turkey would also qualify as a spike diet. It was only fed once, and the diet prior to feeding quail was also only known for 2 days. Turkey reads are contained in the data table, but only at subsamples from days 20 and 27. Is a contamination issue the reason for this or was the proportion of turkey consumed not high enough for stable detection? This itself would be an interesting result.

Lines 119-120: How often were the enclosures cleaned, were there any measures to avoid the contamination of fresh scat with old ones?

Lines 137-228: Laboratory analysis and bioinformatics are very well described.

Lines 193-194: Sentence is not complete, please rephrase.

Line 200: different citation style, please change.

Line 214: missing space

Line 215: different citation style, please change.

Line 233: does “missing” in the negative control mean “no reads at all” or “less than 10 reads”?

Line 260: “amount of each prey species”: please clarify whether absolute prey quantity or percentage of total prey consumption was used in the modelling process. If the latter is the case, please replace the “kg” abbreviations in the model formula.

Table 1: sometimes fist letters of prey items are in lower case.

Figure 1 is not necessary for a better understanding of the general results and could in my opinion be removed from the manuscript.

Lines 297-307: please add a Table containing the model (posterior means, 95% credible intervals significance etc.) in addition to Figure 2.

Line 300 & 301: “prey/kilogram”; “prey per kilogram” please clarify if this is per kg fed to the cheetahs or per cheetah body weight.

Lines 306-307: what is a “high detectability success”? Positive detection in a fresh collected scat?

Lines 324-329: how were these results obtained?

Figure 3: please add the N for each of the species.

Lines 336-338: how was this result obtained?

Figure 4: please explain the light grey dots and the black circles and provide the N.

Lines 336-341: How was this result obtained?

Lines 367-372: please add references on the bias introduced by frequency of occurrence data.

Lines 380-383: as the current study included data on weather conditions it would be great to discuss whether the weather conditions during the experiment were favourable or not for DNA degradation and link this to other results from the literature (e.g. Oehm et al.).

Lines 390-403: Could the amount of bones contained in portions of different prey species have had an effect?

Lines 426-430: This is an interesting result. Was it obtained based on prey only fed once within 3 days? I would suggest using proportion of daily prey consumption instead of an absolute number of 300g. Additionally: This might not be a general result if the defecation rate increases along with an increase in daily consumption.

6. PLOS authors have the option to publish the peer review history of their article (what does this mean?). If published, this will include your full peer review and any attached files.

Reviewer #1: No

---

## [Author Response · Author response to Decision Letter 0]

25 Oct 2019

Plosone: Response to Reviewers

Thuo et al., 2019 “Food from faeces: evaluating the efficacy of scat DNA metabarcoding in dietary analyses” (manuscript PONE-D-19-20752)

Our manuscript was positively received, and the reviewer raised two main concerns and suggested some minor changes to improve the manuscript, which we have fully addressed as outlined below:

Major comments 

1) The authors make a great case for why this experiment was necessary and how future field studies will benefit from the obtained results. Feeding experiments on large mammals are always hard to carry out because of limited individuals available and specific requirements to minimize stress of the animals. My main concern with this manuscript is twofold: on the one hand, materials and methods and the raw data uploaded as supporting information do not contain all the information necessary to fully comprehend the analyses. On the other hand, I honestly doubt that the feeding regime applied in this experiment permits all described analyses and conclusions. That being said, I would like to emphasize that such a trial is definitely useful prior to a large field sampling campaign even though it does not necessarily permit statistically robust answers to all questions raised in the introduction.

Response: We appreciate the reviewer’s comment and agree that future field studies will benefit from the results obtained in this experiment. We concur with reviewer’s comment on the possible limitations in feeding experiments of large mammals and have emphasised in our discussion that the results should be carefully interpreted. Below, we have addressed each of the main concerns raised by the reviewer with particular emphasis on the materials and methods as well as ensuring the complete raw data is available. In addition, we have provided more detailed explanation why we believe our feeding regime presents the best estimates of the likely outcomes. 

2) Supporting Information

a. Unfortunately, the manuscript and the Supporting Information do not enable the reader to combine data on the consumed diet with the prey detections. Especially for a situation where diet three days prior to defecation and prey detection up to 60 days after defecation plays a role, it would be great to have this information together in one dataset. Additionally, a legend describing the info in the Supplementary file columns would be very useful.

Response: Thank you for raising these points. Indeed, the raw data uploaded as supporting information was unclear since we had initially provided the summarised metabarcoding data and the feeding metadata as separate files. To address this concern, we have concatenated the feeding data with the summarized metabarcoding data into one dataset (see Supplementary file 1). The revised dataset now includes a legend describing all the columns. For reproducibility, we have also provided the summarised metabarcoding data and the feeding metadata as independent files (Supplementary file 2 and Supplementary file 3). We have also provided the R code used to analyse the feeding dataset in the Dryad Digital Repository (https://doi.org/10.5061/dryad.2z34tmpgs ).

b. Some entries in the supplementary table are missing; for example, quail C.day 3, D.day 1 and 3. I am assuming that these were removed because of quail contaminations in the negative controls?! It would be great to see which samples had to be removed and why (contamination, scat consumed by other animals).

Response: Thank you for pointing this out. Some of the entries in the supplementary table are missing because of two reasons: 1. If the scat was eaten or removed most likely by other animals during degradation period and, 2. If the sample produced no reads or was not assigned after bioinformatics process. For clarity, we have added a descriptive text regarding contaminations in line 291-294: “Two of the extraction controls that had shown positive amplification did not result in assignment during ECOTAG process possibly because the initial positive amplification was due to the 12SV5 primers amplifying non-target (e.g. microbial) DNA or due to primer dimer formations”. In addition, we have changed the dataset’s column headers “C.day” and “D.day” into “Collection.day” and “Degradation.day” respectively (see Supplementary file 1).

c. The dataset also does not clearly indicate which part of it was used in the final analysis, and which was not.

Response: For clarity of this section, we have highlighted (in red) the part of the data that was not used in the final analysis and added a note in the dataset legend (see Supplementary file 1).

3. Statistics Analyses:

a. As the two cheetahs were offered different total amounts of prey, it would make more sense to use proportions (e.g. quail was 38% of total daily consumption) in all statistical analyses compared to absolute amount of prey consumed. The wording of the manuscript is not always clear; proportions are displayed in Table 1 but often in the text “amount” is used and in line 248 “kg” is mentioned as a unit.

Response: We have changed the absolute amounts to proportions both in the manuscript text and in all our statistical analysis. No differences were noted in our statistical results after the adjustments.

b. Results and Materials and Methods section do not fit well together i.e. it is not explained in Materials and Methods, how Figures 3 and 4 and the corresponding text were derived. My understanding is that Figure 3 and 4 and the corresponding text are based on detections in fresh scat samples (N = 26), but this might not be the case. In my honest opinion, the analysis presented in the second part of the results section (Figure 3 and 4) are not appropriate under the feeding regime applied in this experiment.

Response: To address this concern we have explained in the Materials and Methods how Fig 3 and 4 (now Figure 2 and 3) and the corresponding text were derived. Fig 2 was based on our finding (see line 229-230, 318-319), that a species is detectable in fresh scats up to three days postfeeding, hence we used our dataset to estimate whether different prey species had different detection probabilities over time and if there is a relationship with the amount eaten by the cheetah (Fig 3). Our results showed that prey detection is influenced by both composition and amount fed to the cheetah’s over time and although there was substantial uncertainty around these estimates, we believe that this is the best likely interpretation based on our experiment. 

Regarding the probability of detection as a function of meal size:

• Based on the feeding regime presented in Table 1, large “meals” (i.e. a prey species consumed in a large quantity) were not switched often. For example, Jura consumed a large meal of deer more than 10 times. Therefore, an increasing detection probability with increasing meal size across days 0 to 3 potentially stems from the same species being fed on consecutive days (or with just one day break) to the same animal.

For analyzing the effect of prey identity and detectability I would suggest using only results from the 26 fresh scats and omit all detections from the dataset where a species was consumed more than once during the 0-3 days time window, thus, reducing the analysis to the rarely consumed prey species like rabbit and quail. This would reduce the analytic power, but at least the nature of the feeding regime would not mask the actual effects.

For analyzing the effect of DNA degradation, I would suggest using scats produced by Jura during a constant “high deer, low chicken” diet and assess the detection probability of both the small meal and the large meal over time.

Response: While we understand the reviewer’s point, we have chosen to use all of the information available to estimate the probability of detecting a prey item in a scat as a function of both diet during the previous three days, and the period of time scats have been left in the open. Rather than using only the fresh scats to assess detectability of prey species, we argue that it is appropriate to use all of the data because detection (or not) of a prey species in a scat that has been left in the open for some time provides relevant information about our ability to detect a prey species. Essentially, we could make a comparison between prey species in their detectability at any time after scats were deposited (e.g. fresh scats, 5 days after deposit, 10 days etc). By including a parameter that models how detectability changes with time since scats were deposited, we are able to statistically correct for any overall decline in detectability over time. This approach substantially increases our ability to detect differences by making full use of the data to estimate both decline over time (which is not large) and detectability differences.

With regards the probability of detection by meal size. We agree that more regular switching of meals would help with estimation of detectability, but we were limited in this by feeding requirements imposed by the zoo. Nevertheless, there was variation in both the composition and amount fed to the cheetahs over time that allowed us to detect prey differences (Figure 2), relationships with the amount eaten (Figure 3), and the effect of days since consumption (Figure 1). We have acknowledged there was substantial uncertainty around most estimates, due to low levels of replication and takes account of the fact that some prey items were not switched often, which then limits our certainty around the effect sizes. But given the constraints of the data, we have presented our best estimates of the likely outcomes. 

Regarding the differences in detectability between prey species:

• For this analysis only, fresh collected scats should be used, as the model presented in Figure 2 shows a negative influence of time since defecation on detection success. If subsamples collected after considerable time are used for this analysis, DNA degradation, environmental conditions and location of the subsample taken from the scat bias detection rates.

Response: As noted in our response above, we believe it is appropriate to use all of the data because detection (or not) of a prey species in a scat that has been left in the open for some time provides relevant information about our ability to detect a prey species. 

• Chicken was very frequently fed to both cheetahs. Therefore, a high detection rate of chicken in scats collected at day 0 after feeding is not surprising, as the chicken signal could easily stem from a meal consumed on the previous day.

Response: We agree that the high detection rate of chicken in scats collected at day 0 could be a result of the high frequency of chicken in the diet prior to data collection. However, we think the result is interesting because deer which was also frequently fed to the cheetahs had a lower detection probability in scats collected at day 0 after feeding. As such, we consider this to be important information for interpreting data from wild collected scats. 

4. The Materials and Methods section or the Results section would benefit from a short description of the contamination levels found in the extraction controls and the amount of detections that had to be removed from the dataset prior to analysis because of contamination.

Response: We have added the following sentence in the results section: “Two of the extraction controls that had shown positive amplification did not result in assignment during ECOTAG process possibly because the initial positive amplification was due to the 12SV5 primers amplifying non-target (e.g. microbial) DNA or due to primer dimer formations.” (line 291 -294)

Minor comments

Comment 1: Line 38: remove “This approach”

Response: In the revised version, the text “This approach” has been deleted (Line 38).

Comment 2: Lines 45-53: To some extent large carnivores are often opportunistic; it would be good to include this aspect here in the introduction.

Response: As suggested, we have added this point in the introduction (Line 52)

Comment 3: Line 62: missing space after efficient.

Response: The missing space after efficient was added (Line 61).

Comment 4: Line 84: missing space between citation numbers

Response: The missing space between citation numbers was corrected.

Comment 5: Line 95, Line 105: “amount consumed” as the two cheetahs were fed different daily amounts, I would suggest changing this into “proportion” and if necessary, changing the statistical analysis accordingly

Response: As suggested we have changed “amount consumed” to “proportion” (line 90 and line 100) and adjusted the same in the manuscript and statistical analysis.

Comment 6: Lines 108-110: is it correct that the less weighting individual was fed larger meals?

Response: That is correct. The overall proportions of meals fed on each cheetah at the Canberra zoo and aquarium is usually adjusted based on their body condition score (Fuller, Meeks, and Dierenfeld 2007; Kellner 2015). Accordingly, during the study period, Innis (weighing less than Jura) was fed on larger meals. For clarity of this section, we have added a reference on meal size calculation for captive cheetahs based on body condition score (line 103)

Comment 7: Line 112: In my opinion, Turkey would also qualify as a spike diet. It was only fed once, and the diet prior to feeding quail was also only known for 2 days. Turkey reads are contained in the data table, but only at subsamples from days 20 and 27. Is a contamination issue the reason for this or was the proportion of turkey consumed not high enough for stable detection? This itself would be an interesting result.

Response: For clarity, turkey did not qualify as a spike diet because it was fed to the cheetah on the first day of the experiment hence, with diet prior being only known for two days. This was contrary to quail which was fed to cheetah on the second day of the experiment and therefore we had records on what the cheetah had been fed 3 days prior (Please see Supplementary file 1 )

Comment 8: Lines 119-120: How often were the enclosures cleaned, were there any measures to avoid the contamination of fresh scat with old ones?

Response: All the cheetah enclosures at the zoo are cleaned daily. To avoid stressing the animals, scat cross-contamination was only avoided by making sure that the enclosures were thoroughly cleaned, and the fresh scats were collected as soon as they were dropped.

Comment 9: Lines 137-228: Laboratory analysis and bioinformatics are very well described.

Response: Authors are grateful to the reviewer for the positive comment.

Comment 10: Lines 193-194: Sentence is not complete, please rephrase.

Response: The sentence has been amended to read “ A total of 209 uniquely labelled libraries from this study (i.e., 193 and 18 libraries originating from scat DNA and negative control samples, respectively) were included in the final superpool “ (line 190).

Comment 11: Line 200: different citation style, please change.

Response: The citation style was adjusted.

Comment 12: Line 214: missing space

Response: The missing space was added.

Comment 13: Line 215: different citation style, please change.

Response: The citation style was corrected.

Comment 14: Line 233: does “missing” in the negative control mean “no reads at all” or “less than 10 reads”?

Response: To clarify the word “missing”, we have added more details in line 229 stating that a species was considered present in a scat subsample if its sequence reads were detected but were missing or less than ten in the corresponding negative control.

Comment 15: Line 260: “amount of each prey species”: please clarify whether absolute prey quantity or percentage of total prey consumption was used in the modelling process. If the latter is the case, please replace the “kg” abbreviations in the model formula.

Response: in the manuscript the “amount of each prey species” was used to mean the absolute prey quantity and had also been used in the model. In the revised manuscript the “amount of each prey species” fed by the cheetah is now expressed as proportion, abbreviation “kg” in the model formula has been changed to “pr”

Comment 16: Table 1: sometimes fist letters of prey items are in lower case.

Response: The first letters of prey items that were in lower case were amended into upper case.

Comment 17: Figure 1 is not necessary for a better understanding of the general results and could in my opinion be removed from the manuscript.

Response: To accommodate this suggestion, Figure 1 was removed from the manuscript.

Comment 18: Lines 297-307: please add a Table containing the model (posterior means, 95% credible intervals significance etc.) in addition to Figure 2.

Response: A table (Table 2) containing the model outputs was added.(between line 312 and 313)

Comment 19: Line 300 & 301: “prey/kilogram”; “prey per kilogram” please clarify if this is per kg fed to the cheetahs or per cheetah body weight.

Response: To improve the clarity of this section, the sentences were rephrased to read “Averaged across all prey species, there was a positive relationship between the probability of detection per kilogram of prey consumed, although this effect was weak on day 0 (the day of consumption), peaked on day 1 (the day after consumption) and then declined in the following two days (Figure 1 and Table 3). (line 297-300)

Comment 20: Lines 306-307: what is a “high detectability success”? Positive detection in a fresh collected scat?

Response: To remove the ambiguity, this sentence has been rephrased to clarify the meaning.

Comment 21:Lines 324-329: how were these results obtained?

Response: To clarify this, our model jointly estimated the probability of detection for each prey species consumed by the cheetah during the previous three days and the time scats were left in the open. In the model S indexes prey species.

Comment 22: Figure 3: please add the N for each of the species.

Response: N for each prey species was added.

Comment 23: Lines 336-338: how was this result obtained?

Response: The result was obtained by calculating the proportion of prey detection (or not) relative to the amount fed.

Comment 24: Figure 4: please explain the light grey dots and the black circles and provide the N.

Response: A sentence in the figure description reading “the grey dots at 0.00 and 1.00 indicate absence or presence of detection of prey items respectively while the black circles shows the proportion of prey detection relative to amount fed” was added to explain the light grey dots and black circles. For clarity the data is derived from the whole dataset. The N has been provided as well.

Comment 25: Lines 336-341: How was this result obtained?

Response: The results were obtained by calculating the proportion of prey detection (or not) relative to the amount fed. For this analysis we used the absolute amount rather than proportions as this analysis aims to estimate the prey detection as a function of absolute amount in this case kilogram eaten.

Comment 26: Lines 367-372: please add references on the bias introduced by frequency of occurrence data. 

Response: As suggested two references were added (line 358) (Klare, Kamler, and MacDonald 2011; Weaver 1993)

Comment 27: Lines 380-383: as the current study included data on weather conditions it would be great to discuss whether the weather conditions during the experiment were favourable or not for DNA degradation and link this to other results from the literature (e.g. Oehm et al.).

Response: To accommodate this suggestion, small changes were made to the structure of lines 375 : “…….which have been shown to reduce prey detection success (Oehm et al., 2011)” (line 375). We have added a reference to this line (Line 375)

Comment 28: Lines 390-403: Could the amount of bones contained in portions of different prey species have had an effect?

Response: This is a valid question, as previous studies have shown that undigested hard part remains (bones, hare etc) in scat samples may play a role in prey DNA detection success, we cannot rule out their potential influence on our results. Since the amount of bones contained in portions of different prey species was not captured in the data, we could not draw conclusions about their effect on detection therefore we have added to the sentence in line 392-393 based on literature: “……or the species had high amount of bones and hair which may have increased their detection rates” suggesting that hard part remains could have potentially influenced our detectability of food DNA in scats, a reference was added (line 390 -391)

Comment 29: Lines 426-430: This is an interesting result. Was it obtained based on prey only fed once within 3 days? I would suggest using proportion of daily prey consumption instead of an absolute number of 300g. Additionally: This might not be a general result if the defecation rate increases along with an increase in daily consumption.

Response: This result was obtained by estimating the probability of detection as a function of amount eaten within 3 days. The absolute amount was changed into proportion. Based on our results, prey is detectable within 3-4 days postfeeding hence an increase in daily consumption may increase the defecation rate but will likely not influence the detection rate within the detection window.

---

## [Editor Report · Decision Letter 1]

13 Nov 2019

Food from faeces: evaluating the efficacy of scat DNA metabarcoding in dietary analyses

PONE-D-19-20752R1

Dear Dr. Thuo,

We are pleased to inform you that your manuscript has been judged scientifically suitable for publication and will be formally accepted for publication once it complies with all outstanding technical requirements.

With kind regards,

Hideyuki Doi

Academic Editor

PLOS ONE

Additional Editor Comments (optional):

I carefully checked the revised manuscript as well as the response letter. I agree the revisions according to the reviewers’ comments and now can recommend to publish the paper in PLOS ONE.
---

## [Editor Report · Acceptance letter]

10 Dec 2019

PONE-D-19-20752R1 

Food from faeces: evaluating the efficacy of scat DNA metabarcoding in dietary analyses 

Dear Dr. Thuo:

I am pleased to inform you that your manuscript has been deemed suitable for publication in PLOS ONE. Congratulations! Your manuscript is now with our production department. 

With kind regards,

on behalf of

Dr. Hideyuki Doi 

Academic Editor

PLOS ONE